# Sex Differences in Colon Cancer: Genomic and Nongenomic Signalling of Oestrogen

**DOI:** 10.3390/genes14122225

**Published:** 2023-12-16

**Authors:** Brian J. Harvey, Harry M. Harvey

**Affiliations:** 1Faculty of Medicine, Royal College of Surgeons in Ireland, RCSI University of Medicine and Health Sciences, D02 YN77 Dublin, Ireland; 2Princess Margaret Cancer Centre, Toronto, ON M5G 1Z5, Canada; harry.harvey@mail.utoronto.ca

**Keywords:** colon cancer, oestrogen, oestrogen receptors, sex differences

## Abstract

Colon cancer (CRC) is a prevalent malignancy that exhibits distinct differences in incidence, prognosis, and treatment responses between males and females. These disparities have long been attributed to hormonal differences, particularly the influence of oestrogen signalling. This review aims to provide a comprehensive analysis of recent advances in our understanding of the molecular mechanisms underlying sex differences in colon cancer and the protective role of membrane and nuclear oestrogen signalling in CRC development, progression, and therapeutic interventions. We discuss the epidemiological and molecular evidence supporting sex differences in colon cancer, followed by an exploration of the impact of oestrogen in CRC through various genomic and nongenomic signalling pathways involving membrane and nuclear oestrogen receptors. Furthermore, we examine the interplay between oestrogen receptors and other signalling pathways, in particular the Wnt/β-catenin proliferative pathway and hypoxia in shaping biological sex differences and oestrogen protective actions in colon cancer. Lastly, we highlight the potential therapeutic implications of targeting oestrogen signalling in the management of colon cancer and propose future research directions to address the current gaps in our understanding of this complex phenomenon.

## 1. Introduction

### 1.1. Sex Differences/Sexual Dimorphism/Gender Differences in Cancer

Sex differences and sexual dimorphism refer to biological and morphological differences, respectively, between males and females in various traits or characteristics. While sexual dimorphism refers to the form or appearances such as height and morphology (evolutionary adaptations), sex differences are primarily associated with biological features, and their impact on cancer is an exciting and rapidly developing area of ongoing research. These terms have been used interchangeably, and not without some misunderstanding, in the scientific literature with reference to male/female differences in cancer [1,2,3]. Gender differences, on the other hand, commonly refer to behavioural and lifestyle traits but are often confused with biological sex differences [4,5,6]. Several types of cancer have demonstrated differences in incidence, presentation, and outcomes between males and females [7,8]. Certain cancers are more prevalent in one sex compared to the other due to anatomical differences, for example, prostate cancer in males, and ovarian or uterine cancer in females, whereas others may be associated with hormonal and genetic factors such as breast cancer [9]. In a survey of the most recent data from the Global Cancer Observatory, it can be noted that the top 20 most common nonreproductive tissue cancers worldwide show sex differences both in incidence and mortality, with females showing lower age-standardized incidence and mortality for all cancers (excluding those of anatomical differences) and for all cancers in every global region surveyed (Figure 1). While hormone and genetic factors contribute to these differences, other nonbiological risk factors can increase the likelihood of incidence and mortality, including smoking and excessive alcohol consumption, particularly for lung, stomach, oesophagus, and liver cancer [10,11]. Here, we discuss the evidence for genomic and nongenomic biological actions of oestrogen underpinning biological sex differences in colon cancer. We focus on more recent reports and try to integrate current knowledge into a holistic understanding of the role of oestrogen, its receptors, and molecular targets, in order to explain this very complex phenomenon for which many mechanistic questions still remain unanswered.

### 1.2. Sex Differences and Regional Variances in Colon Cancer

Colon cancer, also known as colorectal cancer (CRC), is one of the most common types of cancer worldwide. The exact prevalence may vary depending on factors such as age, gender, geographical location, and lifestyle choices. According to the World Health Organization, colon cancer is the second-most commonly diagnosed cancer globally and the third-leading cause of cancer-related deaths (Figure 1). Sex differences showing a female advantage are is a feature of CRC [12]. Although the main risk factor for CRC is age (about 90% of patients are older than 50 years), sex differences in CRC are evident in age-matched male and female patients [13]. The age-standardized mortality rate for men is 50% higher (10.8 per 100,000 person-year) than for women (7.2 per 100,000 person-year) [14].

The CRC incidence rates can differ significantly between countries, but all countries show a male predominance in both incidence and mortality of CRC (Figure 2) [15]. Moreover, the trends for the next 20 years predict a continuing sex difference with more males than females affected by CRC both in its incidence and mortality (Globocan 2020, https://gco.iarc.fr, accessed on 29 July 2023) [16]. Certain risk factors can increase the likelihood of developing colon cancer, including a family history of the disease, a personal history of inflammatory bowel disease, a sedentary lifestyle, a diet high in processed meats and fat and low in fruits, vegetables, and fibre, smoking, and excessive alcohol consumption. It is important to note that advancements in early detection and improved treatment options have contributed to higher survival rates and better outcomes for individuals diagnosed with colon cancer. Regular screenings such as colonoscopies can help detect precancerous polyps or early-stage colon cancer, improving the chances of successful treatment. Biological sex differences, however, are too often a neglected factor in both clinical trials and treatment of colon cancer, although its added value to personalized medicine is incontestable [17]. Sex differences exist at multiple levels in colon cancer, and women have a lower risk of developing CRC than men. Females at a younger age are less likely to die from CRC than age-matched male patients, and certain types of CRC occur predominantly in women. The biological differences in CRC mortality are noticeable in the survival advantage of women during premenopause (18–44 y) compared to men of the same age or to older women post-menopause [18]. These premenopausal advantages and the observations that hormone replacement therapy (HRT) may also be protective in CRC [19,20,21] indicate a role for the female sex steroid hormone oestrogen in delaying the onset and reducing mortality in females with colon cancer [22,23]. Sexual disparity has also been shown in epidemiological studies to be an important factor in the site of onset and metastases in CRC [24]. Women show a higher frequency of right-sided tumours (proximal colon) than men who present more commonly with left-sided (distal colon) tumours [25]. 

Right-sided CRC tumours arise from the ascending colon and proximal two-thirds of the transverse colon, whereas left-sided CRC tumours arise from the descending and sigmoid colon and distal one-third of the transverse colon. The regional origin of proximal (right) or distal (left) colon in CRC can impact prognosis, with CRC of proximal origin being associated with a worse prognosis [26]. Right-sided proximal tumours are harder to detect and diagnose [27] and tend to be more advanced, larger, and poorly differentiated at first diagnosis [28]. Moreover, right-sided proximal tumours tend to be more aggressive and resistant to chemotherapy, which may confound the advantages conferred by oestrogen protection. These regional and sex differences in CRC prognosis may result from differences in cellular molecular subtypes between proximal and distal colon [29]. 

One highly significant candidate is the Wnt signalling pathway and its regional variation along the length of the intestine. There is evidence for Wnt signalling gradients along the intestinal tract and region-specific differences in Wnt responsiveness in CRC [30]. Differential Wnt gene expression profiles and signalling potential have been demonstrated along the colonic crypt–villus axis and throughout the length of the colon, particularly following β-catenin stabilization [31]. This regional variation in Wnt signalling could influence tumour susceptibility between the proximal and distal colon. Genes in the Wnt signalling pathway are more enriched in male compared to female CRC colonic cells and their enhanced sensitization in males may be related to the higher risk and lower survival rates observed in colon cancer in males [32]. Sex differences in CRC patient survival with a female advantage exist for the expression of the Wnt receptor gene *FZD1*. Thus, sex differences and differential expression between proximal and distal colon in Wnt receptor expression and signalling may influence its potential as a therapeutic target [33] Although proximal tumours are more frequent in females, the oestrogen-induced quiescence of Wnt/β-catenin signalling in female CRC [12] will favour normal growth, morphology, and epithelial cell differentiation, whereas a lower activation threshold for Wnt signalling in male CRC will promote cell proliferation, EMT, and tumorigenesis. 

### 1.3. Oestrogen and Sex Differences in Colon Cancer 

Oestrogen is the main sex hormone that controls physiological functions of the female reproductive system, as well as the development of secondary sexual characteristics during female puberty. The predominant circulating oestrogen in humans is 17β-estradiol (E2), which is the most physiologically relevant oestrogen during the female reproductive years. In human females during the reproductive years, the plasma levels of E2 fluctuate over the oestrous cycle, reaching peak concentrations in the follicular phase; 150 to 300 pg/mL (0.5–1 nM) 24 h before ovulation [34], while the highest likely E2 level in the plasma during pregnancy is 8 ng/mL (30 nM) [35]. Oestrogen plasma levels fall off dramatically post-menopause (>45 y), decreasing to below 30 pg/mL (110 pM), which is close to levels found in age-matched males (10–50 pg/mL). Oestrogen at subnanomolar concentrations has been shown to exert multiple sexually distinct physiological actions, with a female advantage in tissues and organs outside the reproductive system, including brain, skeleton, muscle, cardiovascular, immune system, intestine, kidney, liver, and pancreas to influence, respectively, memory, bone strength, cardiac, and skeletal muscle contractility, immunity, intestinal secretion, renal Na^+^ handling, blood pressure, and metabolism [36]. These pleiotropic effects of oestrogen are thought to underpin sex differences and reinforce female physiology, reproduction, longevity, and healthy aging [37]. The beneficial biological effects of oestrogen are lost and often only become apparent after the menopause [38] and may be restored with hormone replacement therapy [39], but only if started early, for example, as in treating cardiovascular disease [40]. Pathophysiological effects of oestrogen are observed in some cancers, notably breast, uterine, and cervical cancers [41], whereas in nonreproductive organ cancers, oestrogen can have protective effects against morbidity and mortality [42]. Oestrogen signalling via specific oestrogen receptors has emerged as a crucial factor in sex differences in the development and progression of colon cancer [12,22,23].

## 2. Oestrogen Receptors in Colon Cancer

Oestrogen receptors determine the biological sex differences, specificity, and transduction of the multiple signalling responses to oestrogen [43]. Probably the most important factor in sex differences in colon cancer development is ligand activation of oestrogen receptors (ER), primarily ERα and ERβ. These receptors are expressed in both males and females but can have differential expression levels and activity patterns, depending on circulating levels of E2, as well as prognostic value in CRC [44]. The role of oestrogen in CRC progression is dependent on the relative abundance of the ER subtypes and their hormone responsiveness. The colonic crypt epithelium expresses both ERα and ERβ [45]. Positional differential expression of ERs has been reported along the colon length and within the crypt axis. ERα is expressed more highly at the base of the crypt of the proximal colon, while ERβ expression is predominant in the midsection of the crypt and in the apical surface cells [46]. This spatial partitioning of ER isoform expression indicates antagonistic roles for ERα and ERβ in transducing differential effects of oestrogen on the physiological function of the epithelial cells located at different sites along the crypt. For example, progenitor cell proliferation at the base of the crypt gives way to enterocyte differentiation in the midsection and shedding of senescent or apoptotic cells at the lumen surface. There is strong evidence that ERβ is more highly expressed in colon cancer tissues from females compared to males, while ERα expression levels can vary among the sexes [44,47]. ERβ has been proposed as a tumour suppressor in CRC, and ERβ Erβ expression is selectively lost during tumour progression through methylation-dependent gene silencing [48]. Thus, the ratio between ERα and ERβ expression and balance in their cell signalling may contribute to sex differences in CRC [49,50]. 

### 2.1. Nuclear Oestrogen Receptors in Colon Cancer

The two main subtypes of ERs are ERα (encoded by oestrogen receptor 1, *ESR1*) and ERβ (encoded by oestrogen receptor 2, *ESR2*), which are differentially expressed in normal colon tissue as well as in colon cancer cells. These two oestrogen receptor subtypes are often referred to as canonical or nuclear ERs, eliciting latent genomic responses to oestrogen and working in an antagonistic fashion on cell biology [51]. The natural ligand for all ERs is the biologically active form of oestrogen, 17β-estradiol (E2). Nuclear ERs play a complex role in colon cancer, influencing various aspects of tumour development and progression [42]. Colon cancer cells expressing ERs can be hormone-responsive, meaning they can respond to oestrogen stimulation. Oestrogen can directly bind to ERs in these cells and trigger a cascade of signalling events leading to changes in gene expression, protein synthesis, and cell behaviour (Figure 3). It is worth repeating that the specific roles and effects of ERs in colon cancer can vary depending on factors such as the subtype of ER (ERα vs. ERβ), their relative expression levels, and the interplay with other signalling pathways. For example, while ERα has been associated with promoting tumour growth and progression, ERβ has been suggested to have a potentially protective or inhibitory effect on colon cancer development [51,52]. Importantly, nuclear ERs are proven prognostic and therapeutic targets in colon cancer [53,54]. 

### 2.2. Membrane Oestrogen Receptors in Colon Cancer

Membrane oestrogen-sensitive receptors have been implicated in CRC [12,23,42], which, unlike the classical nuclear ERs, involve oestrogen-liganded membrane-initiated cell signalling pathways (Figure 3) [55]. Membrane-initiated oestrogen responses in colonic epithelium are typified by rapid nongenomic actions on protein kinase cell signalling pathways, intracellular Ca^2+^ activity, and ion channel function [56], which distinguish these events from the more latent genomic responses to oestrogen in a wide range of normal and cancerous cell types. The presence and role of membrane oestrogen receptors in colon cancer, and cancer in general, are a hot topic of ongoing research and have been the subject of dedicated international meetings continuously since 1992 (RRSH, FASEB SRC Meetings) [57,58,59]. The two most well-characterized membrane receptors for oestrogen are membrane ERα (mERα]) [60] and G protein-coupled oestrogen receptor 1 (GPER1), also known as GPR30 [61], which are discussed in detail below. 

## 3. Genomic and Nongenomic Oestrogen Signalling Pathways in Colon Cancer

Steroid hormone signalling occurs via nongenomic membrane-initiated pathways and genomic nuclear transcriptional pathways working in a coordinated fashion (Figure 3) [62,63]. Oestrogen can trigger cellular signalling in both healthy and cancerous colonic epithelial cells via genomic and nongenomic mechanisms [4,14,42] in a sexually differentiated manner, which also shows dependency on the stage of the oestrous cycle [64]. Both genomic and nongenomic oestrogen signalling pathways in the colon are cell-type-specific and may differ among healthy and cancerous colonic tissues. Genomic oestrogen signalling pathways are characterized by long latency (hours/days) and involve nuclear ERs that bind oestrogen in the cytosol which then dimerize and translocate to the nucleus to bind to specific DNA sites to trigger synthesis of proteins which regulate cell growth, differentiation, and proliferation [65]. Nongenomic oestrogen signalling, on the other hand, is characterized by rapid onset of cell signalling responses (s/min) involving oestrogen binding to a membrane receptor, which, through phosphorylation reactions, can transactivate other membrane receptors and trigger a myriad of cellular signalling pathways impacting upon ion channels, protein kinases, and transcription factors to extend its rapid actions into genomic events [42,66]. The genomic and nongenomic receptor signalling pathways of oestrogen in colonic crypt cells are summarized in Figure 3, showing both protective and exacerbating arms of these pathways in colon cancer. 

### 3.1. Genomic Mechanisms of Oestrogen Signalling in Colon Cancer

The genomic actions of oestrogen in colon cancer are not as well studied compared to its effects on other hormone-responsive cancers like breast or ovarian cancer. While oestrogen receptors have been found in the colon and rectal tissue, the specific role and mechanisms of oestrogen and ERs in colon cancer development and progression between the sexes are still not well understood. Several studies, however, over the past 20 years have suggested potential genomic actions of oestrogen in colon cancer that could underpin sex differences: oestrogen may promote or inhibit cell proliferation and reduce or stimulate cell death (apoptosis) depending on the ER subtype expression in colon cancer cells (Figure 3) [67]. E2–ERα interactions can activate signalling pathways, such as the PI3K/Akt pathway, leading to increased CRC cell growth and survival [68]. Oestrogen has been implicated in promoting angiogenesis, the formation of new blood vessels, which is essential for tumour growth and metastasis [69] and can upregulate the expression of vascular endothelial growth factor (VEGF), a key factor involved in angiogenesis [70]. Oestrogen may also influence epigenetic modifications in colon cancer cells [71] to effect post-translational DNA methylation patterns, histone modifications, and chromatin remodelling, potentially impacting gene expression and cellular behaviour [72]. Nuclear oestrogen receptors may also play a role in inducing or influencing the process of epithelial–mesenchymal transition (EMT) [73], which is involved in tumour metastases in colon cancer [74]. Oestrogen signalling through ERα can regulate the expression of genes associated with EMT, leading to increased cell motility and invasiveness in prostate and breast cancer cells [75,76] and in CRC [12]. Oestrogen signalling has also been found to impact the function and activation of immune cells, including macrophages and T-cells, and regulate the production of cytokines and other immune-related molecules [77] that can affect the tumour microenvironment and immune surveillance against cancer cells [78]. In this regard, oestrogen via ERβ has been shown to modulate the immunogenicity of the tumour microenvironment and immune responses in colon cancer [79]. 

### 3.2. Nongenomic Mechanisms of Oestrogen Signalling in Colon Cancer

It has been known for over two decades that oestrogen can exert sexually segregated rapid nongenomic actions independent of gene transcription on cell signalling in colon [80]. Nongenomic actions of oestrogen in CRC involve signalling pathways that do not require changes in gene expression or protein synthesis [12,23]. Although the nongenomic actions of oestrogen in colon cancer are less well studied compared to its genomic actions, here, we outline some of the potential mechanisms (Figure 3). Activation of membrane receptors: Oestrogen binding to ERα or GPER can trigger intracellular signalling cascades, including activation of protein kinase pathways, calcium mobilization, and stimulation of cyclic adenosine monophosphate (cAMP) production [81]. Rapid kinase activation: Oestrogen can rapidly (s/min) activate various kinases, such as mitogen-activated protein kinases (MAPKs), including extracellular signal-regulated kinase (ERK), c-Jun N-terminal kinase (JNK), and p38 MAPK [82]. These kinases play crucial roles in cell proliferation, survival, and migration, which are important in colon cancer progression [83]. Ion channels: Oestrogen modulates the activity and expression of ion channels in colon cancer cells, such as calcium and potassium channels, leading to changes in intracellular ion concentrations and membrane potential which can affect cell proliferation, apoptosis, and migration [84,85,86]. Activation of second messenger systems: Oestrogen can stimulate the production of second messengers, including cyclic adenosine monophosphate (cAMP) cyclic guanosine monophosphate (cGMP) and inositol trisphosphate (IP3) [87]. These signalling intermediates can regulate various cellular processes, including gene expression, protein phosphorylation, and intracellular calcium release [88]. It is important to note that both genomic and nongenomic actions of oestrogen can contribute to the overall effects of oestrogen on colon cancer cells. The balance between these actions and the interplay with other signalling pathways determine the ultimate impact of oestrogen on colon cancer development and progression.

### 3.3. Cooperativity between Genomic and Nongenomic Oestrogen Signalling in Colon Cancer

Multiple studies in a wide variety of tissues (brain, vascular, epithelial) have shown that genomic and nongenomic cellular responses to oestrogen are not mutually exclusive and can cooperate to produce synergistic effects in physiological functions and in cancer biology (Figure 3) [89]. In this context, it is important to note that a strict dichotomy does not exist between membrane nongenomic and nuclear genomic actions of oestrogen, and evidence exists for cross-talk and integration of these rapid and latent pathways to amplify biological responses to oestrogen [90]. Nongenomic actions of oestrogen have been shown to be both permissive and potentiating for genomic responses [91]. This type of cross-talk can be rapidly initiated by oestrogen actions through membrane ERs to transactivate growth factor receptor tyrosine kinases (EGF and IGF-I receptors) [92], which can produce rapid activation of ERK MAPK and phosphorylation of cytosolic ER to allow its translocation into the nucleus [93] and also cause phosphorylation and recruitment of coactivators (AP-1, Sp-1) to the nuclear transcriptome to amplify the genomic response [94]. Oestrogen signalling through membrane oestrogen receptors can also involve activation (phosphorylation) of protein kinase targets to induce gene transcription and latent nuclear transcriptional activity via ERK, MAPK, c-fos, PI3K/AKT, CREB, and JNK [95,96]. Conversely, genomic actions may amplify nongenomic responses to oestrogen in a sex-specific manner, for example, by stimulating the transcription of protein kinase intermediates of the membrane-initiated signalling pathways in colonocytes [97]. Thus, nuclear ERs are necessary for the expression of proteins that transduce oestrogen effects at the membrane [98]. An important caveat, therefore, in the debate on genomic versus nongenomic oestrogen responses is the accumulating evidence that these are not standalone independent events but show integration and cooperativity and, if considered separately, do not encompass the full range of oestrogen actions in physiology nor in cancer. Another important principle is that labelling nongenomic steroid hormone responses as being solely “rapid” no longer holds true, as latent genomic events may be initiated by earlier membrane–cell signalling [99,100]. Thus, nongenomic and genomic actions of oestrogen can be integrated in targeting cell proliferation pathways in colon cancer biology. 

## 4. Oestrogen Signalling via ERα and ERβ in Colon Cancer 

When oestrogen binds to ERα in the cytosol, the hormone receptor complex dimerizes and translocates to the nucleus where it interacts with specific DNA sequences, oestrogen response elements (ERE), or non-ERE transcription factors, such as c-Jun and c-Fos of the activating protein-1 complex (AP-1), and transcription factor specificity protein 1 (SP1) and NFκB, which extend E2–ERα cell proliferation and proinflammatory actions in colon cancer cells [101]. ERα activation can also initiate signalling pathways that promote cell cycle progression, such as the PI3K/Akt and MAPK/ERK pathways [102]. These pathways, in turn, can stimulate cell growth and survival. From these studies, we can reasonably conclude that oestrogen signal transduction via ERα is protumorigenic in CRC (Figure 3). 

In contrast, oestrogen signalling through ERβ produces antitumorigenic cell signalling responses in a broad range of cancers including CRC by repressing ERα transcription and activating antiproliferative cell signalling pathways (Figure 3) [103]. ERβ upon binding to its ligand oestrogen, dimerizes and translocates to the nucleus where the E2–ERβ complex transcriptionally upregulates target genes which, unlike E2–ERα, promote proapoptotic and antiproliferative responses in CRC. The E2–ERβ complex binds to DNA elements such as ERE or AP1 which activate the FOXO3a gene. Activated FOXO3a in turn transcriptionally upregulates PUMA, p21, and p27, which have been shown to induce apoptosis of CRC cells [104]. ERβ also inhibits the expression of cell proliferating genes such as *c-Myc* and *p45Skp2* [105]. Moreover, some of the EMT and metastasis genes, such as *β-catenin*, *Slug,* and *Twist*, are inhibited by ERβ [106]. Nuclear ERβ can also exert proapoptotic responses in colon cancer cells through increased Caspase-3 activity [107] and inhibit cell proliferation by locking the cell cycle in G1-S phase [108]. There is evidence that ERβ may also inhibit colon cancer cell growth through autophagy mediated by suppression of the mammalian target of rapamycin (mTOR) through Cyclin-D1 degradation [109,110]. In addition to these antitumorigenic actions, E2–ERβ may have immunosuppressive effects in CRC [111] as well as preserving an epithelial phenotype through stimulating the expression of tight junction proteins occludin-1 and JAMA [112] and inhibiting dedifferentiation and epithelial–mesenchymal transition via upregulation of E-cadherin and α-catenin while inhibiting β-catenin [113]. E2–ERβ may also induce microRNA protection by repression of the oncogenic prospero homebox 1 (PROX1) through the upregulation of miR-205, which inhibits colon cancer cell migration and invasiveness [114]. A protective role of ERβ in reducing colon crypt proliferation and inflammatory responses resulting from a high-fat diet has recently been demonstrated both in male and female mice [115].

There is evidence for cross-talk between the ER subtypes, the most important for CRC being the inhibition of ERα transcription by ERβ with oestrogen binding [116] and the negative regulation of the expression of ERα by ERβ [117] as colonocytes differentiate to a mature tight-junction epithelium [118]. Increased ERβ expression could also lead to ERα–ERβ heterodimerization, thereby skewing the expression pattern of target genes from proliferative, antiapoptotic towards an antiproliferative, proapoptotic, and antitumorigenic profile [46]. Disruption of ERβ expression, but not of ERα, increased intestinal neoplasia and promoted tumourigenesis in APC-/- mice [119]. A combined high ERβ expression together with negative ERα expression was found to be correlated with a better prognosis for CRC patients [120]. ERβ expression predominates over ERα in the normal healthy colon and in the initial stages of adenocarcinoma, with progressive loss of ERβ and increased ERα expression observed in colon biopsies in later stages of tumour development [38]. Moreover, the loss of ERβ is associated with enhanced CRC proliferation potential [121], leading to the hypothesis that high ERβ expression may not only be protective against developing CRC but also a prognostic marker and molecular target in the treatment of colon cancer [122,123]. Indeed, a predominant expression of ERβ may show sexual differentiation for its protective role in the early stages of CRC [124].

### 4.1. Nonligand Activation of Nuclear ERs in Colon Cancer

In addition to ligand-binding activation, nuclear ERs can be phosphorylated and transactivated by EGFR activated tyrosine kinases without requiring binding to oestrogen. For example, EGFR can activate the Ras/Raf/MAPK pathway, which phosphorylates ERs, resulting in dimerization and ligand-independent activation of target gene expression [125]. Moreover, the membrane HER2 receptor (erbB-2 receptor tyrosine kinase 2) when bound to EGFR can also activate tyrosine kinases signalling pathways RAS/RAF/ERK, PIK3K/AKT/mTOR, JAK/STAT3 producing a hyperactivation of mitogenic signals leading to uncontrolled cell proliferation and tumorigenesis in CRC [126]. Currently, there are no studies reported in the literature directly linking nonligand activation of ERs to EGFR/MAPK signalling. However, ligand (estradiol) activation of membrane and nuclear ERα has been shown to be linked to EGFR/MAPK signalling to drive proliferation, invasion, and angiogenesis in CRC [42,127].

### 4.2. Membrane Oestrogen Receptors mERα and mERβ in Colon Cancer

While ERα is traditionally recognized as a nuclear receptor that regulates gene expression, detailed evidence accumulated over the past 20 years has demonstrated that a small percentage (approximately 1%) of ERα can also be localized to the cell membrane in certain contexts to produce physiological and clinically relevant biological responses [128]. The membrane ERα (mERα) was first discovered in 1999 [129] and characterized in breast cancer cells by Levin and colleagues [130,131]. The molecular identity was shown to be a palmitoylated variant of the full-length (66 kDa) nuclear ERα receptor which allowed its tethering at the cell membrane in a caveolin-1 signalosome [132,133]. The understanding of the role of the mERα in physiology and cancer biology was hampered by the lack of specific inhibitors and agonists which could distinguish E2–mERα ligand binding from nuclear ERα signal transduction [134]. The nongenomic and rapid actions (s/min) of oestrogen on protein kinases, intracellular calcium, and ion channel activity in colonic crypts [135,136,137] distinguish it from canonical genomic E2–ERα signal transduction which shows typical long latency of hours/days to generate transcriptional responses [138]. It is important to note that the rapid membrane-initiated responses to oestrogen can be elicited at physiological subnanomolar concentrations of oestrogen such that dose–response is of little value in distinguishing nongenomic from nuclear E2–ERα actions [139]. One way to overcome this difficulty is the use of oestrogen analogues which penetrate poorly, if at all, the cell membrane, or the generation of nuclear excluded ERα mutants [140]. In this regard, certain specific membrane-impeded analogues of oestrogen such as E2–BSA [141] and oestrogen dendrimer conjugates (EDC) [142], which do not enter the cytosol to bind nuclear ERα, have been shown to replicate the rapid nongenomic actions of free unbound oestrogen (17β-estradiol). The most significant advance, however, in understanding the physiological and pathological roles of mERα have resulted from the generation by the Levin group of selective mouse models with membrane-only mERα (MOER) or nuclear-only ERα (NOER) expression [143]. Mutations of the palmitoylation site of ERα have also provided a useful tool to dissect membrane-initiated and nuclear actions of oestrogen [144]. These studies have shown the absolute requirement for mERα expression in transducing rapid actions of oestrogen on protein kinases and tyrosine kinase signalling pathways in cancer cell proliferation but not in the development of reproductive organs and tissues [143,144,145]. The presence of ERα on the cell membrane, in addition to its primarily nuclear localization, expands the range of oestrogen actions through combined nongenomic and genomic regulation of cell differentiation and proliferation through an expanded signalosome [146,147].

Regarding the role of membrane ERα in colon cancer, research in this specific area is still evolving, and the understanding of its implications is not yet fully established. However, many of the signalling pathways regulated by membrane oestrogen actions in colon (EGFR, ERK-MAPK, PI3K-Akt, and Wnt) are involved in cell proliferation, survival, and migration in colon cancer cells (Figure 3) [12]. For example, activation of membrane ERα in colonic epithelial cells isolated from females has been associated with the rapid activation of mitogenic signalling cascades, including the MAPK pathway, PI3K/Akt pathway, and Src kinase signalling [80,88,148,149], which can promote cell growth and survival. Additionally, membrane ERα has been implicated in modulating epithelial–mesenchymal transition in breast [150] and colon cancer cells [151], a process involved in CRC tumour invasion and metastasis [152]. 

The current data indicate that mERα is the primary endogenous ER mediator of rapid E2 responses, although a membrane ERβ (mERβ) is also found to be co-expressed with mERα in cancer cells to regulate cell proliferation [153]. The role of mERβ is less well studied in cancer biology, although ERβ may be present at the cell membrane in a palmitoylated form in colon cancer cells to inhibit cell proliferation [154]. Studies have demonstrated that mERβ can produce rapid nongenomic actions of oestrogen on ERK and JNK kinase activity when expressed in Chinese hamster ovarian cells [155], while other studies have shown E2–mERβ to rapidly activate p38 MAPK in human colon cancer cells [156]. The membrane ER subtypes appear to mimic their respective nuclear ER responses to oestrogen, i.e., proproliferative, protumorigenic effects via nuclear ERα and mERα while conversely producing antiproliferative, antitumorigenic effects via both nuclear ERβ and mERβ. In this way, oestrogen nongenomic interactions with membrane ERα and membrane ERβ can modulate cell proliferation, apoptotic pathways, and cell death in CRC [157]. 

### 4.3. Oestrogen Signalling via Truncated ERs in Colon Cancer 

Several splice variants of full-length ERs have been reported in various healthy tissues and cancers [158,159]. These truncated ERα and ERβ receptor proteins arise from mutations in *ESR1* and *ESR2* genes but cannot form homodimers or recruit cofactors like full-length ERs [160]. Truncated ERs may form inactive heterodimers with full-length ERs [161] and collaborate with other oestrogen receptors such as GPER [162] to modulate proliferative and inflammatory responses in cancer. The best studied truncated ER has been ERα36 (36 kDa) in breast and gastric cancers [163,164], which transduces estrogenic nongenomic signalling to promote cell proliferation and metastatic potential [165] Apart from one study reporting decreased ERα36 mRNA expression with advanced Stage CRC [166], the relevance of truncated ERs to oestrogen signalling and sex differences in CRC is unknown and merits further investigation. 

## 5. Oestrogen Signalling via G Protein-Coupled Oestrogen Receptor in Colon Cancer

G protein-coupled oestrogen receptor (GPER), also known as GPR30, first reported in 2005 [167], is a seven-transmembrane receptor that mediates membrane-initiated oestrogen signalling in a wide range of tissues [168]. Initially identified as an alternative oestrogen receptor in breast cancer [169], GPER signalling has emerged as a significant player in colon cancer [170], contributing to various aspects of tumour progression and estrogenic cell signalling responses [171]. While many studies over the past twenty years support an oestrogen-ligand receptor role for GPER [172,173], the involvement of GPER in transducing nongenomic actions of oestrogen in vivo has been challenged [174,175]. Several recent studies, however, have demonstrated the functional expression of GPER in colon cancer cells, which may vary among different colon cancer subtypes and individual tumours [176]. The role of GPER in CRC has garnered much interest as its expression predominates in colon cancer after the loss of nuclear ER, in particular after the loss of ERβ, which has been reported to negatively regulate GPER expression in breast cancer [177]. While the oestrogen protective effects in CRC have been mainly attributed to ERβ, its expression is lost during CRC progression, and this raises the possibility for a role in sex differences of GPER, which remains expressed after ERβ loss in CRC [178]. Chronic mucosal inflammation has been proposed as a precursor of CRC and it is interesting to note that GPER expression shows sex dependence in inflammatory bowel disease (IBD) [179] and may transduce oestrogen protective effects on mucosal barrier function in IBD [180]. The potential therapeutic implications of targeting GPER signalling in cancer have been recently reviewed. This includes the use of GPER agonists or antagonists, alone or in combination with other therapies [181]. 

GPER has been reported to have both tumorigenic and antitumorigenic roles in cancer progression [182,183]. Some studies have shown a protective role for GPER in CRC [184] and its activation has been reported to inhibit cell proliferation in CRC cell lines [185]. Furthermore, a low expression of GPER in CRC was associated with poor patient survival [171]. In contrast, other studies indicate GPER to be protumorigenic in CRC via oestrogen activation by steroid sulfatase [186] and to stimulate cell proliferation in CRC cell lines not expressing nuclear ERs [171]. In support of this hypothesis, GPER expression was shown to be upregulated and stimulated by MAPK signalling in mycotoxin-induced growth of colon cancer cells [187]. In addition, GPER can promote chromosomal instability in CRC, leading to neoplastic transformation and tumour development [188]. The impact of GPER signalling on angiogenesis may also be determinant in its tumour promoter actions [189,190]. GPER may have differential protective or exacerbating effects on CRC tumourigenesis, depending on the expression of ER and activation of cell proliferation signalling pathways.

### Hypoxia and GPER Signalling in Colon Cancer

There is evidence that the divergent roles of GPER in CRC as being either protective or tumorigenic are dependent on the stage of the cancer and the level of hypoxia in the tumour microenvironment and on sex-specific factors influencing hypoxic signals [184]. Moreover, the actions of GPER in the regulation of vascular endothelial growth factor (VEGF) and hypoxia-inducible factor 1-α (HIF-1α) in CRC show sex dependence (Figure 3) [178].

Hypoxia is a key factor in promoting tumour growth through cell signalling involving HIF-1α [191] and its target VEGF, which are associated with poor clinical outcomes in CRC [192]. In a detailed study of the functional consequences of oestrogen actions within the hypoxic CRC cell microenvironment, Bustos et al. [178] found the pro- and antitumorigenic potential of GPER in CRC cell lines to be dependent on the level of oxygen exposure. Under normoxic conditions, oestrogen and the GPER agonist G1 both suppressed CRC cell proliferation. Under hypoxic conditions, however, GPER activation produced the opposite functional effect, with both oestrogen and G1 enhancing CRC cell proliferation, whereas the GPER antagonist G15 inhibited proliferation. Oestrogen treatment enhanced the hypoxia-induced expression of HIF-1α and VEGF, but repressed HIF-1α and VEGF expression under normoxic conditions. The expression or repression of VEGF by oestrogen were mediated by a GPER-dependent mechanism. Thus, GPER is essential in transducing the normoxic antiproliferative effects of oestrogen as well as its hypoxic proliferative actions in CRC cells. The latter response may be amplified by upregulation of GPER expression following exposure to hypoxia and oestrogen. Another protumorigenic amplification factor in the GPER response is the oestrogen-modulated gene, Ataxia Telangiectasia Mutated (*ATM*), which was shown to be repressed in hypoxia via GPER signalling [178]. Loss of *ATM* expression is associated with poor survival in CRC [193] and an increase in phosphorylated ATM protein levels has been observed in hypoxic colon cancer cells [194]. The modulation of *ATM* expression by GPER in low oxygen tension and the sensitivity of its expression to oestrogen in CRC provides an additional mechanism for protumorigenic actions of oestrogen via GPER in colon cancer under hypoxic conditions. Thus, it is important to take into account the CRC stage and tumour microenvironment when interpreting the role of E2–GPER interactions in colon cancer tumorigenesis, sex differences, and patient survival/treatment [170].

The involvement of GPER in CRC patient survival displays clear sex differences. In a cohort of 566 CRC patient tumour samples, *GPER* expression was significantly associated with poor survival in CRC Stages 3–4 females but not in the stage-matched male population [178]. Since a hypoxic tumour microenvironment is associated with late stages in CRC, we may conclude that sex differences in this case are underpinned by E2–GPER tumorigenic actions on HIF-1α/VEGF activation and on ATM suppression under hypoxia. The antiproliferative effects of E2-GPER signalling in normoxia may explain the observations of protective effects of GPER expression on CRC survival in the early stages of cancer development. 

## 6. Oestrogen Regulation of Wnt/β-Catenin Signalling in Colon Cancer 

Wnt/β-catenin signalling plays a key role in various biological processes, including embryonic development, tissue homeostasis, and cell proliferation [195,196]. Dysregulation of the Wnt/β-catenin signalling pathway has been strongly associated with the development and progression of colon cancer [197]. The Wnt/β-catenin signalling pathway is commonly hyperactivated in colon cancer and plays a crucial role in CRC development and progression [198]. Oestrogen has been shown to modulate this pathway in a sex-dependent manner in CRC [199] and in breast and endometrial cancers [200,201] through reciprocal interaction between ERα and β-catenin. In reproductive tissues, oestrogen promotes cell proliferation via ERα stimulation of β-catenin nuclear translocation [202]. In contrast, other studies indicate that ERα is inhibitory for Wnt/β-catenin-mediated proliferation and neoplasia in nonreproductive tissues, for example, in liver cancer [203]. Moreover, ERβ was reported in the latter study to have no role in oestrogen modulation of Wnt/β-catenin signalling. These studies suggest that oestrogen may have an inhibitory effect on Wnt/β-catenin signalling in nonreproductive tissue cancers in females, potentially contributing to sex differences in CRC incidence and outcomes. But how does this work in CRC and what are the factors that determine whether oestrogen will exert a protective or exacerbating role via Wnt/β-catenin pathway signalling in CRC, in particular when the expression of ER is lost with advancing tumorigenesis? To answer this question, we must first understand the possibility of cross-talk with other receptors and signalling pathways. The Wnt signalling pathway interacts with various signalling pathways, particularly K^+^ channels, implicated in colon cancer. Interaction between the Wnt/β-catenin pathway and the oestrogen-regulated K^+^ ion channel KCNQ1 in CRC is a major molecular mechanism that displays sex differences in the regulation of CRC cell proliferation and epithelial–mesenchymal transition [12]. 

### 6.1. Oestrogen Regulation of Wnt-KCNQ1 Interactions in Colon Cancer

KCNQ1 is a voltage-gated K^+^ channel (Kv7.1) expressed in the basolateral membranes of colonic crypts where it functions to provide the electrical driving force for Cl^−^ secretion [204]. In colon, the KCNQ1 channel is coexpressed with the β-regulatory subunit KCNE3, which greatly increases the ionic conductance of the channel and confers voltage and cAMP sensitivity to the channel [205]. KCNQ1 has been shown to have an antitumorigenic role in many gastrointestinal (GI) cancers, including colon cancer [206]. Moreover, relapse-free CRC patient survival was positively associated with high *KCNQ1* gene expression, which displayed sex-dependence for a female advantage [207]. KCNQ1 channels modulate Wnt signalling in a wide number of GI cancers [208] and there is strong evidence for bidirectional interaction between KCNQ1 and β-catenin in normal healthy colon and in colon cancer cells [207]. Oestrogen regulates KCNQ1:KCNE3 channel function by uncoupling KCNQ1 from KCNE3 via PKCδ-dependent phosphorylation of KCNE3 at residue Ser82, which destabilizes the KCNE3:KCNQ1 channel complex [209]. The uncoupling of KCNQ1 from KCNE3 promotes KCNQ1 endocytosis and recycling in colonic crypts [210], which allow KCNQ1 to leave the plasma membrane and bind cytosolic activated β-catenin. KCNQ1 then returns β-catenin to the cell membrane to trap it in a complex with E-cadherin at adherens junctions (Figure 4) [207]. In this way, KCNQ1 anchors β-catenin at the plasma membrane, stabilizing adherens junctions, and promoting cell–cell adhesion. The association of KCNQ1 with β-catenin at the cell membrane is essential to preserve a well-differentiated epithelial phenotype by maintaining a stable KCNQ1: β-catenin:E-cadherin complex at adherens junctions and preventing epithelial–mesenchymal transition [211]. Trapping β-catenin with KCNQ1 at the cell membrane has been shown to retard the nuclear translocation of β-catenin and prevent the transcriptional activation of proliferative genes [207]. These observations suggest that oestrogen-regulated KCNQ1 channel quells the Wnt:β-catenin nuclear signalling pathway to suppress CRC cell proliferation and EMT, but only in females [212].

### 6.2. Oestrogen Regulation of Wnt Receptor Oncogenic Signalling in Colon Cancer

The receptor tyrosine kinase pathway, particularly the EGFR pathway, can cross-regulate Wnt signalling in colon cancer [213]. Additionally, the Transforming Growth Factor-β (TGF-β) pathway can modulate Wnt signalling through Smad proteins [214], all of which could amplify the cell proliferation responses in CRC. In contrast, a recent study has provided evidence for a role of GPER in protecting against CRC progression by selectively reducing the oncogenic effects of hyperactive Wnt/β-catenin signalling pathways in CRC [215]. Firstly, sex differences were observed in the gene expression of the Wnt receptor *FZD1* (Frizzled 1) in Kaplan–Meier survival analyses across multiple CRC patient gene microarray datasets. High expression of *FZD1* was associated with poor relapse-free survival rates in the male but not in the female CRC population. Secondly, activation of GPER with the G1 agonist prevented the Wnt-pathway-induced upregulation of the *JUN* oncogene. These novel findings indicate a mechanistic role for GPER in protecting against CRC progression in females by selectively reducing the tumorigenic effects of Wnt/β-catenin oncogenic signalling pathways. 

## 7. Oestrogen Regulation of Epigenetic, Microbiome and Metabolic Factors in Colon Cancer

Other emerging factors underpinning sex differences in CRC include microRNAs and the gut microbiota. MicroRNAs (miRNAs) are small noncoding RNA molecules that regulate gene expression by targeting specific messenger RNAs for degradation or translational repression. Differential expression of miRNAs has been observed between males and females in CRC [216,217]. Oestrogen can modulate the expression and activity of specific miRNAs, which in turn influence the expression of genes involved in CRC development [218]. These sex-specific miRNA profiles may contribute to the oestrogen-mediated sex differences in CRC. This has been shown for miR-30c-5p expression which was associated with better survival in females and was downregulated in males [219]. Several miRNAs are currently under investigation for therapeutic applications in CRC [220].

Recent studies suggest that the gut microbiota can influence CRC development and responses to oestrogen [221]. The composition and function of the gut microbiota can differ between males and females [222], potentially affecting oestrogen metabolism and signalling. Thus, another contributory factor to sex disparities in CRC is the oestrogen–gut microbiome axis [223]. Microbial metabolites produced by the gut microbiota may modulate oestrogen receptor activity and alter oestrogen levels, contributing to sexual differentiation in CRC [224]. In addition, E2 may alter the gut microbiota to reduce the risk of developing CRC [225].

Another potential factor contributing to CRC sex differences is oestrogen metabolism. The metabolism of oestrogen is an important process that can influence its bioavailability and activity. In females, oestrogen is predominantly metabolized by cytochrome P450 enzymes, including CYP1A1 and CYP1B1, which convert estradiol to 2-hydroxyestradiol (2-OHE2) [226]. In males, oestrogen is mainly metabolized by CYP1A2 and CYP3A4, leading to the formation of 4-hydroxyestradiol (4-OHE2) [227]. The different metabolic pathways can result in distinct oestrogen metabolite profiles, which may contribute to sex variances in CRC susceptibility. There is some evidence that testosterone can exert neoplastic actions in CRC [227] and strong evidence for the presence of upregulated genes on the Y chromosome that contribute to colorectal cancer in males by driving tumour invasion and aiding immune escape [41,228,229].

## 8. Comparison of Oestrogen and Testosterone Signalling in Colon Cancer 

Over 90% of CRC cases are sporadic, arising from transition from normal mucosa to adenoma, followed by the development of carcinoma usually involving mutations in antiproliferative genes, which include *APC* (adenomatous polyposis coli), and proliferative tumorigenic genes, such as *K-ras* and *p53.* CRC is not solely a hormone-dependent cancer compared to predominantly oestrogen- or androgen-driven cancers such as breast or prostate, respectively. However, oestrogen and testosterone have been implicated in modulating the initial adenoma and mesenchymal transitions through effects on tumour promoter genes, and on inflammatory, apoptosis, and angiogenesis signalling pathways.

Sex-based prevalence and mortality of colon cancer have been observed in many animal studies where ovariectomized rats and mice are more prone to develop CRC while castrated male animals have a lower risk in developing CRC in response to mutations in the *Apc* tumour-suppressor gene [230]. These studies indicate that female sex hormones are actively protective in CRC whereas male sex hormones can actively promote CRC [231]. More precisely, oestrogen protection appears to be conferred through cell signalling via ERβ receptors [232], while the nuclear androgen receptor A (AR-A) transduces the exacerbating effects of testosterone [233]. The B-isoform of the nuclear AR (AR-B) may confer protection in CRC as its loss is associated with adenoma formation, although the membrane-associated AR (mAR) may also confer protection in CRC [234]. These protective/exacerbating effects of oestrogen or androgens in CRC are also observed in human subjects supplemented with oestrogen therapy [235] or testosterone [236]; there is debate, however, on the exact role of testosterone in CRC [237]. Although the precise oestrogen and testosterone signalling pathways underpinning sex differences in CRC still remain elusive, we can make certain observations supported by the literature and highlight testable hypotheses on potential pathways that differentiate males and females in CRC outcomes, as summarized in schematic form in Figure 5. 

Oestrogen may offer protection in females through switching off the Wnt/β-catenin signalling pathway, thereby reducing nuclear translocation of β-catenin (by promoting its localization at adherens junctions) with its transcription factor TCF-4 [201] and suppressing the activation of proliferative genes such as *JUN*, *K-ras*, *CTNNB1.Myc*, *PROX1*, and *P35* [215,238,239,240,241]. Oestrogen can also reduce the potential for activation of Wnt signalling by reducing the expression of the WNT receptor FZD1 [215]. Moreover, oestrogen, by inactivating Wnt/β-catenin, may synergise with APC to negatively regulate canonical Wnt signalling and stabilization of β-catenin to counteract cell proliferation, invasion, and EMT by promoting epithelial differentiation and apoptosis, thereby suppressing tumour progression [242]. If some β-catenin:TCF-4 complex does escape into the nucleus, APC could further antagonize Wnt signalling by counteracting β-catenin in the nucleus. Oestrogen also has anti-inflammatory actions which may combat chronic inflammation, which, in some cases, can be a precursor of CRC. Oestrogen has been shown to reduce the activation of proinflammatory cytokines IL-1, IL-2, IL-6, IL-8, and monocyte chemoattractant protein-1 (MCP-1), as well as suppressing inflammatory transcription factor NFκB and inhibiting Tumour Necrosis Factor-α (TNF-*α*) [243,244,245,246,247]. Moreover, oestrogen activates anti-inflammatory IL-4 and IL-1 and prevents monocyte and neutrophil invasion [248]. Oestrogen also reinforces the immune system by activating Interferon-γ (IFN-γ) [249], which plays a key role in the activation of cellular immunity and the stimulation of antitumour immune-responses via helper T-cells and B-cells. In addition, the double XX chromosome also confers approximately 20% more immune genes to females [250] and reinforces the expression of immune genes [251] such as the X-linked immune gene *TLR7* [252]. Finally, oestrogen inhibition of HIF-1α and VEGF in normoxia may confer a female advantage in early CRC stages by suppressing the initial transition from normal mucosa to adenoma [178]. 

In males, testosterone exerts pro-tumorigenic actions by stimulating the expression of proliferative genes and proinflammatory mediators that promote cell growth, cell survival [253], epithelial–mesenchymal transition, and cell invasiveness [254]. Moreover, it is unlikely that oestrogen can promote antitumorigenic actions in males due to the low circulating oestrogen levels and reduced oestrogen receptor expression. Estradiol produces a negative feedback inhibition of luteinizing hormone secretion from the pituitary, and subsequently testosterone, release [255], thus further lowering the production of oestrogen via aromatase activity in the adrenals and adipose tissue. Therefore, contrary to females, oestrogen may confer little or no protection against CRC in males [256].

## 9. Conclusions and Perspectives

An increasing number of epidemiological and molecular studies have demonstrated a female sex advantage in colon cancer such that the incidence. morbidity and mortality of CRC is lower in premenopausal females compared to males and postmenopausal females. The protective effect of oestrogen against CRC has been proposed as a possible explanation for this observation.

Membrane oestrogen receptors, particularly mERα, GPER, and nuclear oestrogen receptor ERα, ERβ, have been detected in colon cancer tissue. The presence of these receptors suggests that oestrogen nongenomic and genomic signalling could have an impact on CRC tumourigenesis in a sex-specific manner. Oestrogen signalling may protect against colon cancer development and growth through its effects on cell proliferation, apoptosis, cell cycle progression, and epithelial differentiation. These beneficial effects of oestrogen are absent in males, with the added disadvantage of testosterone and Y chromosome tumorigenic actions in CRC (Figure 5). It is worth emphasizing that in some cases, oestrogen signalling might be detrimental in CRC, as in a hypoxic tumour microenvironment which is a hallmark of late-stage CRC. 

Oestrogen receptor status in CRC tumours may serve as a prognostic marker, providing information on the likelihood of disease progression and patient outcomes. ER-positive tumours in females might be associated with a better prognosis compared to ER-negative tumours. Some colon cancers that express oestrogen receptors may be candidates for targeted hormonal therapies. Selective oestrogen receptor modulators or aromatase inhibitors, which are used in hormone-receptor-positive breast cancer, have been explored in preclinical studies and early-phase clinical trials for colon cancer [257]. It is essential to note that not all colon cancers express oestrogen receptors, and the clinical relevance of oestrogen signalling targets may vary between individuals and tumour subtypes [258]. Therefore, identifying patients who may benefit from hormone-based therapies requires careful patient selection and further research.

## Figures and Tables

**Figure 1 genes-14-02225-f001:**
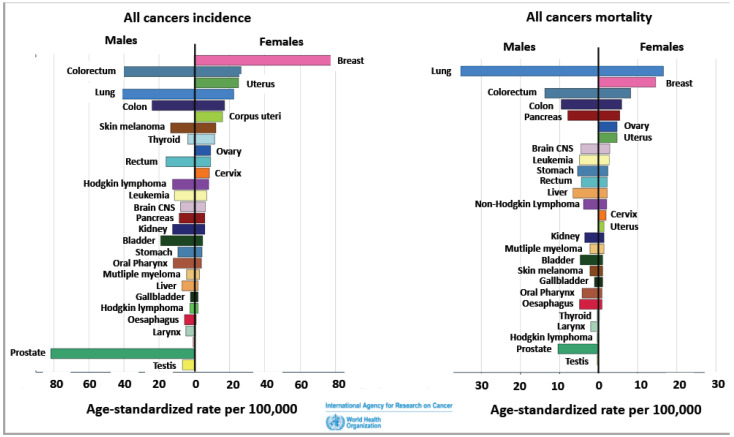
Age-standardized incidence and mortality rates for all cancers per 100,000 EU, USA, and Canada. Survey of most recent data from the Global Cancer Observatory Globocan 2020, https://gco.iarc.fr/overtime/en/dataviz/bars?sexes=1_2&sort_by=value2&mode=cancer, (accessed on 29 July 2023).

**Figure 2 genes-14-02225-f002:**
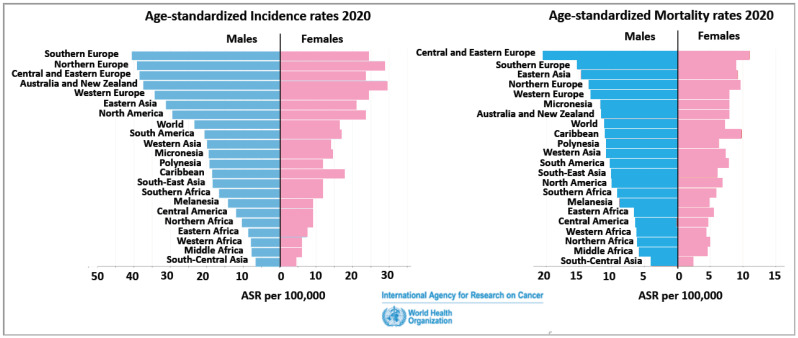
Age-standardized incidence and mortality rates for colorectal cancer per 100,000 across WHO regions. Survey of most recent data from the Global Cancer Observatory, Colon Cancer 1989–2012. https://gco.iarc.fr/overtime/en/dataviz/bars?sexes=1_2&sort_by=value2&cancers=5&types=1 (accessed on 29 July 2023).

**Figure 3 genes-14-02225-f003:**
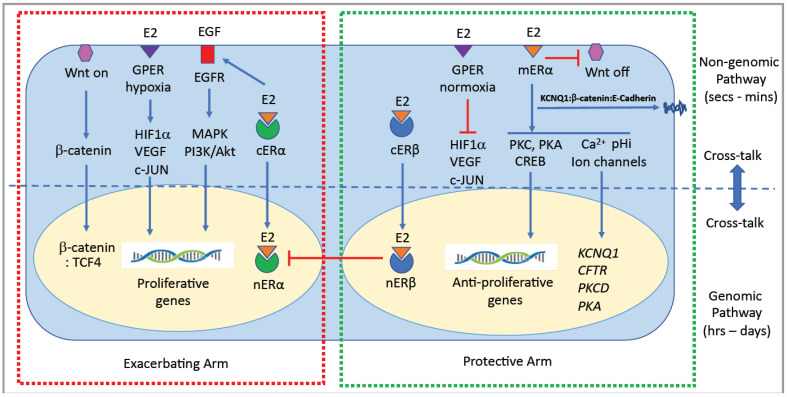
Genomic and non-genomic estrogen (E2) signaling pathways in colon cancer. In the genomic pathway, estrogen interacts with estrogen receptors in the cytosol (cERα and cERβ) which translocate to the nucleus to activate (nERα) or repress (nERβ) cell proliferative genes. ERβ can also inhibit ERα nuclear transcription. Non-genomic to genomic cross-talk can occur when ERα transactivates EGFR to initiate cell proliferative signalling via MAPK, PI3K/Akt. Membrane estrogen receptors mERα and GPER signal through non-genomic pathways to activate protein kinases, calcium mobilization or intracellular pH, which in turn modulate ion channel activity or activate transcription factors such as CREB or HIF-1α and VEGF which can in turn cross-talk to genomic signaling pathways to increase the synthesis of protein kinases or ion channels. GPER activation can have dual effects on CRC cell proliferation depending on oxygen levels in the tumour microenvironment. In normoxia, E2-GPER inhibits HIF-1α, VEGF and c-Jun transcription of proliferative genes whereas in hypoxia E2-GPER activates these HIF-1α proliferative pathways. In addition Wnt/β-catenin proliferative pathways can be inhibited by the estrogen regulated ion channel KCNQ1 to trap β-catenin at adherens junctions and impede nuclear translocation of β-catenin. Red and green squares encompass signaling pathways which exacerbate CRC or protect against CRC, respectively.

**Figure 4 genes-14-02225-f004:**
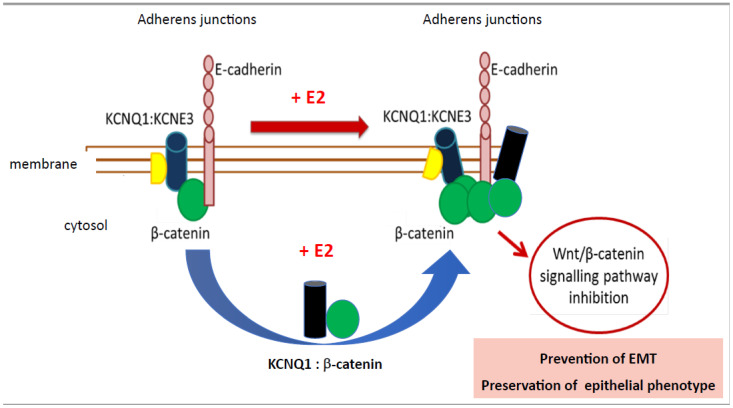
KCNQ1 anchors β-catenin at the plasma membrane, stabilizing adherens junctions and promoting cell–cell adhesion to maintain a highly differentiated, tight junction epithelial phenotype in healthy colon. Oestrogen uncouples KCNQ1 from KCNE3, allowing the channel protein to enter an endocytosis recycling pathway and capture activated β-catenin in the cytosol, returning it to the cell membrane in association with E-cadherin. The trapping of β-catenin at the plasma membrane prevents its nuclear translocation and suppresses the Wnt pathway activation of cell proliferation genes as well as inhibiting epithelial–mesenchymal transition. Adapted from References [201,203,204].

**Figure 5 genes-14-02225-f005:**
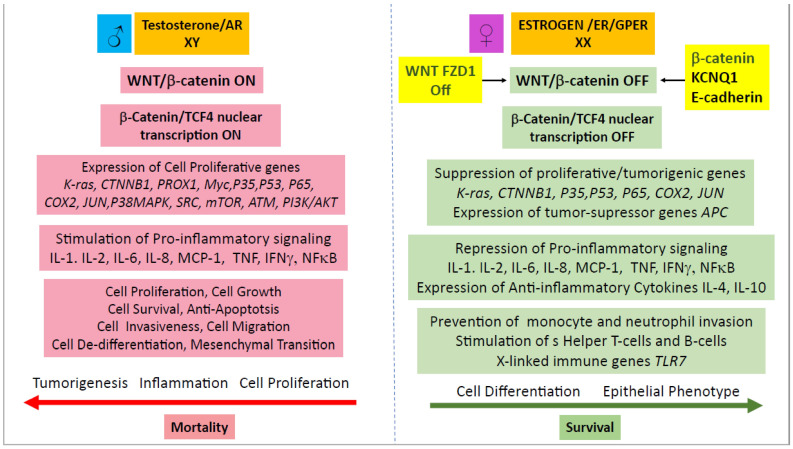
Schematic representation of advantages for oestrogen and disadvantages for testosterone cell signalling mechanisms underpinning sex differences between females and males in colon cancer. In females, oestrogen inactivates the Wnt/β-catenin nuclear transcription pathway to suppress the expression of proliferative genes while allowing the expression of antiproliferative genes. Oestrogen also reinforces mucosal immunity while supressing inflammatory mediators. These oestrogen-induced antitumorigenic actions are permissive in maintaining a well-differentiated epithelial phenotype and enhancing CRC patient survival. In males, testosterone activates the expression of proliferative genes and proinflammatory mediators to promote cell growth, invasiveness, and epithelial–mesenchymal transition, leading to an enhanced tumorigenic potential and higher risk of mortality.

## Data Availability

Not applicable.

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
