# Peer review of "Sex Differences in Colon Cancer: Genomic and Nongenomic Signalling of Oestrogen"

_genes, 2023, doi:10.3390/genes14122225_

Round 1

Reviewer 1 Report

Comments and Suggestions for Authors

Harvey and Harvey wrote a review on Sex differences in colon cancer, highliting different aspects. The manuscript is well written and structured in paragraphs, but there are things to improve. Improve the quality of the figures and above all cut and merge some subsections (from section 2 to 6) because they are redundant and repetitive. For these reasons I suggest these revisions.

Author Response

We thank the Reviewers for taking the time to consider our manuscript and offer helpful comments and suggestions to improve the Review. We note that all Reviewers consider the manuscript well-written and should be published.

Referee 1

Comments and Suggestions for Authors

Harvey and Harvey wrote a review on Sex differences in colon cancer, highlighting different aspects. The manuscript is well written and structured in paragraphs, but there are things to improve. Improve the quality of the figures and above all cut and merge some subsections (from section 2 to 6) because they are redundant and repetitive. For these reasons I suggest these revisions.

Authors Reply:

Thank you for the suggestions to improve the quality of the Review paper.  Perhaps the Figure images appear blurred in the submitted manuscript as these were imbedded into the Word file from high resolution TIFF files and this type of file transfer into Word docs always reduces the image quality. The original high resolution TIFF files of the Figures are now uploaded with the revised manuscript. Some sections have been merged (shown in marked-up text) to reduce redundancy and repetition.

Reviewer 2 Report

Comments and Suggestions for Authors

This review explores in detail the role of estrogen in colorectal cancer. It summarises and clearly defines sex specific differences in non-reproductive cancers and the broad range of evidence that females are protected against colorectal cancer in particular. The review then focuses on estrogen, the nuclear and membrane receptors and both the genomic and non-genomic pathways activated by estrogen. The authors explore interactions between these pathways and other well characterised signalling pathways such as WNT, AKT, Hypoxia and EMT.

                Their arguments are logical and well supported and display a wide range of knowledge across the field. The often contrasting findings in the literature are laid out, usually with good explanations for the disparity. The figures summarise some of the key mechanisms well.

Overall this is an interesting review and should be published. There are a few minor corrections needed:

·         Figures 1 and 2 need to be reproduced at higher quality with larger axis labels

·         Figure 1 has both colorectal and colon cancer bars which are probably overlapping and not well defined

·         There are some typos, for example on page 12, the β (beta) symbol is missing in multiple places

Discussion points:

·         The authors discuss the fact that the site of tumourigenesis varies between male and female, with a female bias towards the proximal colon. They also describe literature showing that estrogen has an inhibitory effect on WNT. Could these two observations be linked? A few studies (eg Leedham et al 2013) show that WNT is higher in the proximal colon perhaps making it more resistant to estrogen induced inhibition?

·         The section on non-ligand activation of nuclear ERs is brief but interesting and could mean that ERα expression/activity in CRC may also impact male patients. Is there any further literature on this or whether estrogen receptor activation correlates with EGFR/MAPK activity?

Author Response

Comments and Suggestions for Authors

This review explores in detail the role of estrogen in colorectal cancer. It summarises and clearly defines sex specific differences in non-reproductive cancers and the broad range of evidence that females are protected against colorectal cancer in particular. The review then focuses on estrogen, the nuclear and membrane receptors and both the genomic and non-genomic pathways activated by estrogen. The authors explore interactions between these pathways and other well characterised signalling pathways such as WNT, AKT, Hypoxia and EMT.

            Their arguments are logical and well supported and display a wide range of knowledge across the field. The often contrasting findings in the literature are laid out, usually with good explanations for the disparity. The figures summarise some of the key mechanisms well.

Overall this is an interesting review and should be published. There are a few minor corrections needed:

Authors Reply:

Thank you for these helpful comments on our review. We reply to each suggestion below:

  • Figures 1 and 2 need to be reproduced at higher quality with larger axis labels

Authors Reply:

Figures 1 and 2 have been edited to provide higher quality definition between cancer types and axis labels.

  • Figure 1 has both colorectal and colon cancer bars which are probably overlapping and not well defined

Authors Reply:

Colorectal and Colon bars now well defined in Figure 1

  • There are some typos, for example on page 12, the β (beta) symbol is missing in multiple places

Authors Reply:

We have checked for typos.  We cannot find typos for the β (beta) symbol on page 12 nor on any other pages. Perhaps the reviewer is using an older version of Microsoft Word which does not display the beta symbol.

Discussion points:

  • The authors discuss the fact that the site of tumorigenesis varies between male and female, with a female bias towards the proximal colon. They also describe literature showing that estrogen has an inhibitory effect on WNT. Could these two observations be linked? A few studies (e.g. Leedham et al 2013) show that WNT is higher in the proximal colon perhaps making it more resistant to estrogen induced inhibition?

Authors Reply:

This is an interesting observation and we have added a few lines in the text highlight the possibility of WNT expression in different locations in the colon and its possible linkage to sex differences in tumorigenesis with reference citations. Marked-up text lines 96-122:

Right-sided CRC tumors arise from the ascending colon, and proximal two thirds of the transverse colon and  left-sided CRC tumors arise from the descending and sigmoid colon, and distal one third of the transverse colon.The regional origin of proximal (right) or distal (left) colon in CRC can impact prognosis, with CRC of proximal origin being associated with a worse prognosis [Mik M, Berut M, Dziki L, Trzcinski R, Dziki A. Right- and left-sided colon cancer - clinical and pathological differences of the disease entity in one organ. Arch Med Sci. 2017;13(1):157–162. doi: 10.5114/aoms.2016.58596]. Right-sided proximal tumours are more frequent in females and are harder to detect [which are harder to detect and diagnose [White A, Ironmonger L, Steele RJC, Ormiston-Smith N, Crawford C, Seims A. A review of sex-related differences in colorectal cancer incidence, screening uptake, routes to diagnosis, cancer stage and survival in the UK. BMC Cancer. 2018 Sep 20;18(1):906. doi: 10.1186/s12885-018-4786-7] and tend to be more advanced, larger and poorly differentiated at first diagnosis [Baran B, Mert Ozupek N, Yerli Tetik N, Acar E, Bekcioglu O, Baskin Y. Difference Between Left-Sided and Right-Sided Colorectal Cancer: A Focused Review of Literature. Gastroenterology Res. 2018 Aug;11(4):264-273. doi: 10.14740/gr1062w]. These regional  and sex differences in CRC prognosis may  result from differences in cellular molecular subtypes between proximal and distal colon [Richman S, Adlard J. Left and right sided large bowel cancer: have significant genetic differences in addition to well-known clinical differences. BMJ Br Med J. 2002;324(7343):931–932. doi: 10.1136/bmj.324.7343.931].

One highly significant candidate is the WNT signaling pathway and its regional variation along the length of the intestine. There is evidence for WNT signaling gradients along the intestinal tract and region-specific differences in WNT responsiveness in CRC. Differential WNT gene expression and signalling potential have been demonstrated along the colonic crypt-villus axis and  increased WNT signalling throughout the length of the colon, particularly following b-catenin stabilization [Leedham SJ, Rodenas-Cuadrado P, Howarth K, Lewis A, Mallappa S, Segditsas S, Davis H, Jeffery R, Rodriguez-Justo M, Keshav S, Travis SP, Graham TA, East J, Clark S, Tomlinson IP. A basal gradient of Wnt and stem-cell number influences regional tumour distribution in human and mouse intestinal tracts. Gut. 2013 Jan;62(1):83-93. doi: 10.1136/gutjnl-2011-301601}. This regional variation in WNT signaling could influence  tumour susceptibility between proximal and distal colon.  Regional differences in WNT pathway gene expression and signaling have been reported in CRC [Adam, R.S., van Neerven, S.M., Pleguezuelos-Manzano, C. et al. Intestinal region-specific Wnt signalling profiles reveal interrelation between cell identity and oncogenic pathway activity in cancer development. Cancer Cell Int 20, 578 (2020). https://doi.org/10.1186/s12935-020-01661-6]. Genes in the WNT signaling pathway are more enriched in males compared to females CRC  and their enhanced sensitization in males may be related to the higher risk and lower survival rates observed in colon cancer in males [Lopes-Ramos CM, Kuijjer ML, Ogino S, Fuchs CS, DeMeo DL, Glass K, Quackenbush J. Gene Regulatory Network Analysis Identifies Sex-Linked Differences in Colon Cancer Drug Metabolism. Cancer Res. 2018 Oct 1;78(19):5538-5547. doi: 10.1158/0008-5472.CAN-18-0454]. Sex differences in CRC patient survival with a female advantage exist for the expression of the WNT receptor gene FZD1 [Ref 210  Abancens M, Harvey BJ, McBryan J. GPER Agonist G1 Prevents Wnt-Induced JUN Upregulation in HT29 Colorectal Cancer Cells. Int J Mol Sci. 2022 Oct 20;23[20]:12581. doi: 10.3390/ijms232012581]. Thus, sex differences and differential expression between proximal and distal colon in WNT receptor expression and signaling  may influence its potential as a therapeutic target [Zhao H, Ming T, Tang S, Ren S, Yang H, Liu M, Tao Q, Xu H. Wnt signaling in colorectal cancer: pathogenic role and therapeutic target. Mol Cancer. 2022 Jul 14;21(1):144. doi: 10.1186/s12943-022-01616-7], [Disoma C, Zhou Y, Li S, Peng J, Xia Z. Wnt/β-catenin signaling in colorectal cancer: Is therapeutic targeting even possible? Biochimie. 2022 Apr;195:39-53. doi: 10.1016/j.biochi.2022.01.009]. Although more aggressive proximal tumours are more frequent in females, the estrogen-induced quiescence of Wnt/b-catenin signaling in female CRC [Ref 12: Abancens M, Bustos V, Harvey H, McBryan J, Harvey BJ. Sexual Dimorphism in Colon Cancer. Front Oncol. 2020 Dec 9;10:607909. doi: 10.3389/fonc.2020.607909.] will favour normal growth, morphology, and epithelial cell differentiation whereas a lower activation threshold for WNT signaling in male CRC will promote cell proliferation, EMT and tumorigenesis.

The section on non-ligand activation of nuclear ERs is brief but interesting and could mean that ERα expression/activity in CRC may also impact male patients. Is there any further literature on this or whether estrogen receptor activation correlates with EGFR/MAPK activity?

Authors Reply:

We reply to this interesting suggestion in new text added at Lines 376-379:

Currently there are no studies reported in the literature directly linking non-ligand activation of ERs to EGFR/MAPK signaling. However, ligand (estradiol) activation of membrane and nuclear ERa has been shown to be linked to EGFR/MAPK signaling to drive proliferation, invasion, and angiogenesis in CRC [Das PK, Saha J, Pillai S, Lam AK, Gopalan V, Islam F. Implications of estrogen and its receptors in colorectal carcinoma. Cancer Med. 2023 Feb;12(4):4367-4379. doi: 10.1002/cam4.5242], [Krasinskas AM. EGFR Signaling in Colorectal Carcinoma. Patholog Res Int. 2011 Feb 14;2011:932932. doi: 10.4061/2011/932932]

Reviewer 3 Report

Comments and Suggestions for Authors The manuscript is well written describing the sex differences in colon cancer. Although it seems to lose interest as it misses to point out clearly the differences in colon cancer based on the signaling of estrogen. Please provide a clear representative figure of the difference in sex in colon cancer under the influence of estrogen signals.    Also, line 628 mentions figure 5 which seems to be missing from the manuscript. 

Author Response

Referee 3

Comments and Suggestions for Authors

The manuscript is well written describing the sex differences in colon cancer. Although it seems to lose interest as it misses to point out clearly the differences in colon cancer based on the signaling of estrogen.

Authors Reply:

The review is all about sex differences in colon cancer underpinned by estrogen cell signaling. Therefore we have difficulty understanding the remark by the Reviewer.

Given that the Reviewer has some trouble in  appreciating our extensive description of  CRC sex differences based on estrogen signalling, we have included a new section to provide added value to our Review by clearly identifying these differences and which we feel will  further strengthen the Review in its clarity and impact. New Section 8 added; Lines 648-705:

8  Comparison of estrogen and testosterone signaling in colon cancer

Over 90 % of CRC cases are sporadic arising from  transition from normal mucosa to adenoma, followed by the development of  carcinoma usually involving mutations in anti-proliferative genes which include APC (adenomatous polyposis coli),  and proliferative tumorigenic genes such as K-ras and p53. CRC is not solely a hormone-dependent cancer compared to predominantly estrogen or androgen driven cancers such as breast or prostate, respectively. However , estrogen and testosterone have been implicated in modulating the initial adenoma and mesenchymal transitions through effects on tumour promoter genes, and on inflammatory,  apoptosis and angiogenesis signaling pathways.

Sex-based prevalence and mortality of colon cancer has been observed in many animal studies where ovariectomised rats and mice are more prone to develop  CRC while castrated male animals have a lower risk in developing CRC in response to mutations in the  Apc tumour-suppressor gene [Amos-Landgraf JM, Heijmans J, Wielenga MCB, Dunkin E, Krentz KJ, Clipson L, et al. Sex disparity in colonic adenoma-genesis involves promotion by male hormones, not protection by female hormones. Proc Natl Acad Sci U S A. 2014;111(46):16514–16519. doi: 10.1073/pnas.1323064111. These studies indicate that female sex hormones are actively protective in CRC whereas mae sex hormones can actively promote CRC [Harbs J, Rinaldi S, Gicquiau A, Keski-Rahkonen P, Mori N, Liu X, Kaaks R, Katzke V, Schulze MB, Agnoli C, Tumino R, Bueno-de-Mesquita B, Crous-Bou M, Sánchez MJ, Aizpurua A, Chirlaque MD, Gurrea AB, Travis RC, Watts EL, Christakoudi S, Tsilidis KK, Weiderpass E, Gunter MJ, Van Guelpen B, Murphy N, Harlid S. Circulating Sex Hormone Levels and Colon Cancer Risk in Men: A Nested Case-Control Study and Meta-Analysis. Cancer Epidemiol Biomarkers Prev. 2022 Apr 1;31(4):793-803. doi: 10.1158/1055-9965.EPI-21-0996.].  More precisely, estrogen protection appears to be conferred through cell signaling via ERb receptors [Slattery ML, Sweeney C, Murtaugh M, Ma KN, Wolff RK, Potter JD, et al. Associations between ERalpha, ERbeta, and AR genotypes and colon and rectal cancer. Cancer Epidemiol Biomark Prev. 2005;14(12):2936–2942. doi: 10.1158/1055-9965.EPI-05-0514.] while the nuclear androgen receptor A (AR-A) transduces the exacerbating effects of testosterone [Catalano MG, Pfeffer U, Raineri M, Ferro P, Curto A, Capuzzi P, et al. Altered expression of androgen-receptor isoforms in human colon-cancer tissues. Int J Cancer. 2000;86(3):325–330. doi: 10.1002/(SICI)1097-0215(20000501)86:3<325::AID-IJC4>3.0.CO;2-G.] . The B-isoform of the nuclear ER (AR-B) may confer protection in CRC as its loss is associated with adenoma formation,  although the membrane-associated AR (mAR) may also confer protection in CRC [Gu S, Papadopoulou N, Nasir O, Föller M, Alevizopoulos K, Lang F, et al. Activation of membrane androgen receptors in colon cancer inhibits the prosurvival signals Akt/Bad in vitro and in vivo and blocks migration via vinculin/actin signaling. Mol Med. 2011;17(1-2):48–58. doi: 10.2119/molmed.2010.00120].  These protective/exacerbating effects of estrogen or androgens in CRC are also observed in human subjects supplemented with estrogen therapy [Chlebowski RT, Wactawski-Wende J, Ritenbaugh C, Hubbell FA, Ascensao J, Rodabough RJ, et al. Estrogen plus progestin and colorectal cancer in postmenopausal women. N Engl J Med. 2004;350(10):991–1004. doi: 10.1056/NEJMoa032071] or testosterone [Amos-Landgraf JM, Heijmans J, Wielenga MCB, Dunkin E, Krentz KJ, Clipson L, et al. Sex disparity in colonic adenoma-genesis involves promotion by male hormones, not protection by female hormones. Proc Natl Acad Sci U S A. 2014;111(46):16514–16519. doi: 10.1073/pnas.1323064111] although there is debate on the exact role of testosterone in CRC [Roshan MH, Tambo A, Pace NP. The role of testosterone in colorectal carcinoma: pathomechanisms and open questions. EPMA J. 2016 Nov 10;7(1):22. doi: 10.1186/s13167-016-0differences in CRC 071-5} Although the precise estrogen and testosterone signaling pathways underpinning sex differences in CRC still remain elusive, we can make certain observations supported by the literature and highlight testable hypotheses on potential pathways as summarised in schematic form in Figure 5.

Estrogen may offer protection in females through switching off the Wnt/b-catenin signaling pathway to reducing nuclear translocation of b-catenin (by promoting its localisation at adherens junctions) with its transcription factor TCF-4  and suppressing the activation of proliferative genes such as JUN, K-ras, CTNNB1 and  P35. Estrogen can also reduce the potential for activation of Wnt signaling by reducing the expression of the WNT receptor FZD1. Moreover estrogen by inactivating Wnt/b-catenin can also synergise with APC to negatively regulate canonical Wnt signaling and counteract cell proliferation, invasion and EMT by promoting epithelial differentiation, apoptosis and suppress tumor progression. If some b-catenin:TCF-4 does escape into the nucleus, APC could further antagonize Wnt signaling by interacting with and counteracting β-catenin in the nucleus. Estrogen also has anti-inflammatory actions and chronic inflammation can be a precursor of CRC. Estrogen reducing the activation of pro-inflammatory cytokines IL-1. IL-2, IL-6, IL-8, monocyte chemoattractant protein-1 (MCP-1),  as well as suppressing inflammatory transcription factor NFkB and inhibiting tumour Necrosis Factor-α (TNF-α). Moreover, estrogen activates anti-inflammatory IL-4, IL-1, and prevents monocyte and neutrophil invasion. Estrogen also reinforces the immune system activating Interferon-γ (IFN-γ) which plays a key role in activation of cellular immunity and subsequently, stimulation of antitumor immune-response via helper T-cells and B-cells. In addition, the double XX chromosome also confers approx. 20% more immune genes to females such as the X-linked immune gene TLR7. Finally, estrogen iinhibition of HIF-1a and VEGF in normoxia confers a female advantage to suppress the initial transition stages from normal mucosa to adenoma.

In males, testosterone exerts pro-tumorigenic actions by stimulating the expression of proliferative genes and pro-inflammatory mediators which promote cell growth, cell survival,

Epithelial mesenchymal transition and cell invasiveness. Moreover, it is unlikely that estrogen can promote anti-tumorigenic actions in males due to the low circulating estrogen levels and reduced estrogen receptor expression.  Estradiol produces a negative feedback  inhibition of  luteinizing hormone secretion from the pituitary and subsequently testosterone release and thus further lowering the production of estrogen via aromatase activity in the adrenals and adipose tissue. Therefore, contrary to females, estrogen may confer little or  no protection against CRC in males [Cohen PG. Aromatase, adiposity, aging and disease. The hypogonadal-metabolic-atherogenic-disease and aging connection. Med Hypotheses. 2001;56(6):702–708. doi: 10.1054/mehy.2000.1169].

Please provide a clear representative figure of the difference in sex in colon cancer under the influence of estrogen signals.

Authors Reply:

A representative figure of the sex differences in colon cancer under the influence of estrogen was provided in the Graphical Abstract. To help the reviewer and readers, we have generated an additional figure (Figure 5) showing in schematic form the various estrogen (and testosterone)  cell signals between males and females which affect patient survival in CRC.

Also, line 628 mentions figure 5 which seems to be missing from the manuscript. 

Authors Reply:

Figure 5 is now included in the revised manuscript.